# Utilization of Olive Oil Processing Waste Composts in Organic Tomato Seedling Production

**Yüksel Tüzel [1,*], Kamil Ekinci [2] , Gölgen Bahar Öztekin [1] , İbrahim Erdal [2] , Nurhan Varol [3] and Özen Merken [4]**

[1] Department of Horticulture, Faculty of Agriculture, Ege University, 35100 Bornova-Izmir, Turkey; golgen.oztekin@ege.edu.tr

[2] Departments of Agricultural Machinery and Technology Engineering and Soil Science and Plant Nutrition, Faculty of Agriculture, Isparta University of Applied Sciences, 32200 Isparta, Turkey; kamilekinci@isparta.edu.tr (K.E.); ibrahimerdal@isparta.edu.tr (İ.E.)

[3] Ministry of Agriculture and Forestry, Olive Research Institute, 35100 Bornova-Izmir, Turkey; varol.nurhan@gmail.com

[4] Viticulture Research Institute, 45125 Yunusemre, Manisa, Turkey; ozen.merken@gthb.gov.tr

* Correspondence: yuksel.tuzel@ege.edu.tr; Tel.: +90-232-3111398

**Abstract:** Olive oil byproducts show differences according to the olive oil extraction systems, which are called olive mill solid wastes, olive oil wastewater and olive oil wastewater sludge. Three different kinds of composts, including two-phase and three-phase olive mill solid wastes, and olive oil wastewater sludge were produced with separated dairy manure, poultry manure, and straw. The composts obtained from two-phase and three-phase olive mill solid wastes and olive oil wastewater sludge were named as two-phase, three-phase, and water sludge composts, respectively. They were separately enriched by rock phosphate and potassium salt. These composts were mixed with peat in a ratio of 0%, 25%, 50%, 75%, and 100% (*v/v*). Tomato seeds were sown in all mixtures on 3 February 2016. All the seeds were sown into 2 trays and each plug included 2 replicates. The trays were left in a germination room for 3 days, then moved to a heated greenhouse which is specialized for growing seedlings, and the seedlings were grown there for 3 weeks. The results showed that increasing compost ratios in the growing medium and also the enrichment of the growing medium increased organic matter content, electrical conductivity, and macro and micro nutrient concentrations. The germination period lasted longer with increasing compost ratios. The shoot length was lower at a compost ratio of over 50% excluding water sludge compost, which reacted to over 75%. The highest plant dry weights were obtained in the plants grown on the media with compost ratios of 50%, 25%, and 25% for water sludge compost, enriched two-phase compost, and enriched three-phase compost, respectively. We concluded that the composts obtained from two-phase and three-phase olive mill solid wastes and olive oil waste water sludge can be used without any need of enrichment and a ratio of 25% was found appropriate in most of the measured properties.

**Keywords:** *Solanum lycopersicum*; olive oil waste; two-phase; three-phase; water sludge

## 1. Introduction

Turkey is one of the most important olive- and olive oil-producing countries among the Mediterranean countries with a production of 1,500,467 tonnes of olive in 2018 [1]. The share of organic olive production among the total production in 2018 was 14.22% [2]. Two-phase or three-phase olive oil processing systems are used for the extraction of olive oil and both systems generate large amounts of by-products, which are called two-phase and three-phase olive mill solid

wastes, olive oil waste water and olive oil wastewater sludge [3]. In Turkey, a survey study showed that all the shares of producers running three-phase, two-phase, and traditional cold stone pressed olive oil production systems were 71%, 27%, and 2%, respectively [4].

Olive oil production produces a large amount of solid and liquid wastes each year. Three-phase olive mill solid wastes contain broken seeds of olive. Olive mill wastewater contains 83%–96% water, 3.5%–15% organic matter, and 0.5%–2.0% mineral salts, depending on factors such as olive varieties, growing conditions, soil and climatic conditions, extraction methods, etc. [5]. Both effluents pose environmental problems since they exhibit highly phytotoxic and antimicrobial properties mainly due to phenols and they are not easily biodegradable [6–8]. Therefore, olive processing wastes have been considered as soil and water pollutants and cannot be used directly for agricultural purposes [7]. Within the framework of the measures taken by the Ministry of Environment and Urbanization of the Republic of Turkey, it is recommended that factory owners accumulate olive mill wastewater in lagoons or open ponds, evaporate their water, and utilize olive oil waste water sludge as the least risky solution for the environment. Additionally, factories should convert their processing systems into a two-phase system. Chowdhury et al. [9] reported that two-phase systems produce a lignocellulosic olive humid husk, which is a watery solid by-product with high contents of water (56.6%–74.5%) and phenols (0.62%–2.39%).

As a result, it is necessary to utilize solid and liquid wastes from both systems. Numerous researchers indicate that composting of olive oil production wastes with manure and some other organic materials is the best way of recycling as agricultural material [10,11]. The composted olive oil processing solid waste can be utilized as organic inputs for soil fertility and plant nutrition in agricultural production.

Fertilization is the most important input necessary for the conservation and maintenance of soil fertility in crop production in organic agriculture. On the other hand, with the growth in agricultural production, the amount of organic wastes arising from agriculture-based industry is increasing day by day. By composting these resources, it is possible to obtain organic raw materials that are beneficial to the soil and to protect the environment [12]. At the same time, rational input can be provided in organic agriculture for plant nutrition. Cegarra et al. [11] stated that the final form of composted olive oil processing solid waste has a higher organic matter content and remarkable mineral elements without toxic elements.

Several studies were carried out on the applications of compost obtained from olive oil processing wastes in agricultural production. Raviv et al. [13] applied composts produced from solid and liquid wastes of olive oil mill on tomato seedlings. Michailides et al. [14] produced compost from three-phase olive pomace waste and olive leaves and tested it on lettuce yield. Killi and Kavdır [15] carried out a study on the effects of compost produced from three-phase pomace waste on tomato yield. Diacono and Montemurro [16] conducted a study on the effects of composts obtained from two-phase olive pomace on the yield of organic emmer crop. However, none of these studies carried out a comparative study as to the effects of composts obtained from two-phase and three-phase olive mill solid wastes and olive oil waste water sludge separately on the growth performance of *Solanum lycopersicum* L. seedlings in growing medium.

The purposes of this study were to evaluate composts obtained from two-phase and three-phase olive mill solid wastes and olive oil waste water sludge, to determine the effects of enrichment of composts, and to compare different compost rates on organic tomato seedling production.

## 2. Materials and Methods

This study was conducted during the years of 2015 and 2016. Composts were produced at Composting Facility in Olive Research Institute, Ministry of Agriculture and Forestry. Then, they were tested in seedling production at the Horticulture Department of the Faculty of Agriculture at Ege University, Izmir, Turkey (38°27′17″ N, 27°14′17″ E). Organically certified seeds of tomato cultivar 'Şencan-9' (provided from Ataturk Central Horticultural Research Institute, Yalova, Turkey) were

used for the study. Compost materials were obtained from the mixture of olive oil processing wastes from two and three phase systems (two-phase and three-phase olive mill solid wastes, olive oil waste water and olive oil waste water sludge) with separated dairy manure, poultry manure, and wheat straws. All input materials were obtained from the organically certified farms. The optimized mixing ratios for 3 different kinds of composts determined at Composting Laboratory at the Department of Agricultural Machinery and Technology Engineering in Isparta University of Applied Sciences (Table 1) were produced based on dry weight (Table 1) were produced (based on dry weight) [17].

**Table 1.** The optimized mixing ratios for 3 different kinds of composts used in this research.

|  | 2P | 3P | WS |
|---|---|---|---|
|  | **Mixing Ratios (%)** | | |
| Two-phase olive mill solid wastes | 60 | - | - |
| Three-phase olive mill solid wastes | - | 46 | - |
| Olive oil waste water |  | 1 | - |
| Olive oil waste water sludge | - | - | 20 |
| Separated dairy manure | 23 | 27 | 53 |
| Poultry manure | 10 | 21 | 21 |
| Wheat straws | 7 | 5 | 6 |
| C/N ratio | 30.17 | 25.26 | 20.16 |

An aerated static pile composting method was used for composting the wastes (Figure 1). Piles with a width of 2 m, a length of 3 m and a height of 1.50 m were formed. Rutgers aeration strategies [18] were performed for aeration of piles for 360 days, which is in agreement with those reported in the study of Chowdhury et al. [9]. Although the composting process was monitored for temperature, pH, electrical conductivity (EC), moisture, and organic matter contents, C, N, and heavy metals ratios, and total phosphorus, they are not reported here. At the later stages, 0.38 kg of cotton seed meal per one kg of dry matter of the initial compost was added to each compost pile to enrich the composts at day 330 of composting (maturation and stabilization stages). Additionally, 0.16 kg of rock phosphate and 0.02 kg of raw potassium salt [19] per one kg of dry matter of the initial compost was added to each compost pile for the enrichment of composts at day 360 of composting. Composting lasted for 425 days including the maturation and stabilization periods. This prolonged period was due to the enrichment (E) process of composts. Therefore, the enriched versions of each compost were labeled as E2P, E3P, and EWS. Powder sulfur was applied at the fourth month of composting to reduce the pH value in the piles. For this purpose, 8 g of powder elemental sulfur was applied to one kg of dry compost [20].

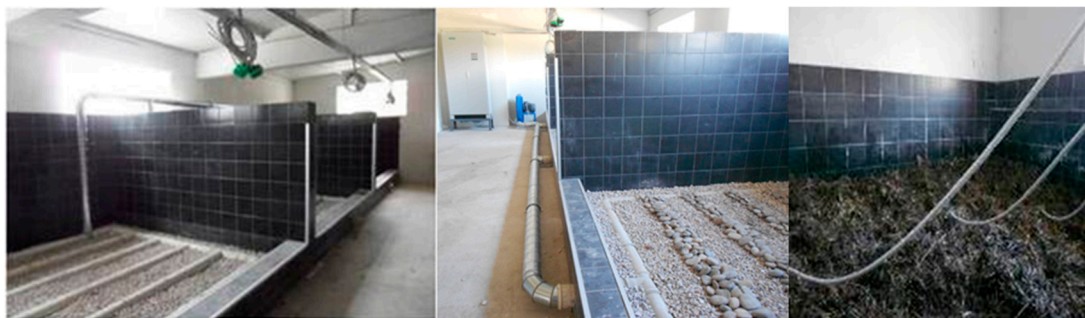

**Figure 1.** Aerated static pile composting system.

Peat provided from Denizli local peat bogs (Turkey) and composts (2P, 3P, WS, E2P, E3P, and EWS) were used as organic substrates in the growing media with compost ratios (%, *v/v*) of 0%, 25%, 50%, 75%, and 100% with local peat. Neither lime nor any nutrient was added into the peat.

Tomato seeds were sown in all growing media on 3 February 2016. All the seeds were sown into 2 trays with 128 plugs in each. Each plug included 2 replicates. After sowing, the trays were left in a germination room at a day/night temperature of 24/24 °C and 80% relative humidity (RH) under dark conditions for 3 days, then moved to a heated greenhouse (15/24 °C and 70%RH) which is specialized for growing seedlings and the seedlings were grown there for 3 weeks. The seedlings were fertilized with liquid farmyard manure (Botanica, Camli Yem Besicilik, Izmir, Turkey) (2 cc L$^{-1}$, EC:1.32 dS m$^{-1}$, pH:4.6) every day with a boom system based on the previous results of Tuzel et al. [21]. In this period, the germination rate and germination period of the seeds were noted. The germination rate was calculated by counting the number of germinated seeds in the cells and expressed as %. The germination period was determined as the number of days required for 50% seed emergence.

When the seedlings were ready for planting in a month, they were harvested from each replicate containing 20 seedlings of treatments in order to measure shoot and root biomass. The roots were washed and cleaned from the growing medium and separated from the shoots. The root and shoot (stem and leaf) samples were weighed for fresh weight (g) and dried for 48 h in a thermo-ventilated oven at 65 °C. Then, these dried samples were weighed for dry weight (g) and dry matter was calculated as (%). The longest root length (from top to bottom) was measured with a tape meter and the average result was expressed in cm. The distance between the starting point of the roots and the tip of plant leaves was measured again with a tape meter (cm) and the values were used as seedling height. Stem diameter was also measured above the root collar of the seedlings between nodium with digital caliper (mm).

Minolta colorimeter (CR-400, Minolta Co., Tokyo, Japan) was used to determine leaf color as CIE L* a* b*. The obtained values of a* and b* were used to calculate hue angle [$h° = \tan^{-1} (b*/a*)$] and chroma [$C* = \sqrt{(a*^2 + b*^2)}$], which determine the saturation and the essential components of the color (red, yellow, blue, and green), respectively [22]. The total chlorophyll index was measured with a chlorophyll meter (SPAD-502Plus, Konica Minolta, Chiyoda-Tokyo, Japan) and expressed as SPAD.

In order to determine plant nutrient concentrations, the seedlings were harvested after the experiment period over the soil surface. Then, they were washed with tap water and distilled water to clean surface residues, dried at 65 °C until constant weight, and were grounded. The samples were wet digested with a microwave digestion system and then filtered up to 50 mL with de-ionized water for P, K, Ca, Mg, Cu, Zn, and Mn measurement. Except for P, other nutrients in the supernatant were measured using an atomic absorption spectrophotometer (AAnalyst 400, Perkin Elmer, Waltham, MA, USA). Phosphorus was determined calorimetrically using the spectrophotometer (TU1880 Double Beam UV-VIS, PG Instruments, Leicestershire, UK). In order to determine the N concentration, the samples were wet digested in 250 mL macro-Kjeldahl tubes using concentrated $H_2SO_4$ and Khjeldahl tablet at 350–400 °C. After digesting the samples with NaOH (40%), $NH_4$-N was fixed in $H_3BO_3$ (2%) and titrated with 0.1 N $H_2SO_4$ [23]. The same procedures and methods were applied to determine the mineral compositions of composts and peat used in the growing media and their mixtures as in plant analysis. The organic matter content of the dry samples of materials was analyzed after incinerating the samples at 550 °C as recommended by the US Department of Agriculture and the US Composting Council [24]. The pH and EC of the fresh samples were extracted by shaking at 180 rpm for 20 min at a solid:water ratio of 1:10 (*w/v*) [25], and were measured using pH (pH 720, WTW, Weilheim, Germany) and EC (Multi 340i, WTW, Weilheim, Germany) meters.

The experimental design was randomized blocks with 4 replicates (*n* = 20). A factorial analysis was performed with the composts (WS, EWS, 2P, E2P, 3P, E3P) and ratios (0%, 25%, 50%, 75%, and 100% with local peat and the interaction between these 2 factors. The data were subjected to analysis of variance to determine any statistically significant differences by using the JMP statistical analysis package program (SAS Institute, Cary, NC, USA). The Tukey range test was conducted at a 5% significance level.

## 3. Results

### 3.1. Physical and Chemical Properties of Substrates

Some physical and chemical characteristics of the seedling growing media were determined before seed sowing (Tables 1–3). The organic matter, content of the media was 38.45% at the initial stage. However, when the compost ratio was increased from 0% to 100% in the growing media at the start, the organic matter contents increased with the rate of 38.49%, 28.32%, 41.40%, 19.25%, 62.21% and 67.70% for WS, EWS, 2P, E2P, 3P and E3P, respectively. The highest organic matter (64.48%) was determined for E3P with a compost ratio of 100%. EC of the local peat was 1.11 dS m$^{-1}$ before seed sowing. By the use of composts, EC values increased dramatically in particular when the composts were enriched and used with 75% and/or more. The pH of the growing media changed between 5.60 and 7.38. The pH decreased with an increasing compost ratio in the growing medium (Table 2).

**Table 2.** Initial organic matter, EC, and pH values of the growing media.

| Composts | Compost Ratios in Peat (%) | Organic Matter (%) | EC (dS m$^{-1}$) | pH |
|---|---|---|---|---|
| WS | 0 | 38.45 h | 1.11 n | 7.38 a |
| | 25 | 42.11 fgh | 1.77 lmn | 7.13 a |
| | 50 | 47.01 d–h | 2.26 k–n | 6.69 abc |
| | 75 | 45.93 d–h | 3.26 f–k | 6.28 bcd |
| | 100 | 53.25 b–e | 4.49 ef | 5.60 d |
| EWS | 0 | 38.45 h | 1.11 n | 7.38 a |
| | 25 | 38.58 h | 2.75 h–m | 6.77 abc |
| | 50 | 44.08 e–h | 3.69 f–i | 7.13 a |
| | 75 | 42.63 fgh | 4.30 efg | 7.01 ab |
| | 100 | 49.34 d–g | 6.23 cd | 6.64 a |
| 2P | 0 | 38.45 h | 1.11 n | 7.38 a |
| | 25 | 38.30 h | 1.65 mn | 7.28 a |
| | 50 | 44.29 e–h | 2.35 j–n | 7.33 a |
| | 75 | 49.34 d–g | 2.19 k–n | 6.92 abc |
| | 100 | 54.37 bcd | 3.59 f–j | 6.16 cd |
| E2P | 0 | 38.45 h | 1.11 n | 7.38 a |
| | 25 | 38.30 h | 2.00 k–n | 6.96 ab |
| | 50 | 42.53 fgh | 2.98 g–l | 7.02 ab |
| | 75 | 48.90 d–g | 4.30 efg | 6.95 ab |
| | 100 | 45.85 d–h | 7.39 c | 7.06 a |
| 3P | 0 | 38.45 h | 1.11 n | 7.38 a |
| | 25 | 40.48 gh | 2.06 k–n | 7.13 a |
| | 50 | 47.57 d–h | 2.99 g–l | 7.06 a |
| | 75 | 48.89 d–g | 2.45 i–m | 7.35 a |
| | 100 | 62.37 ab | 3.79 e–h | 6.78 abc |
| E3P | 0 | 38.45 h | 1.11 n | 7.38 a |
| | 25 | 43.71 e–h | 3.67 f–j | 7.19 a |
| | 50 | 50.95 c–f | 5.04 de | 7.33 a |
| | 75 | 59.80 abc | 9.08 b | 7.25 a |
| | 100 | 64.48 a | 11.14 a | 7.04 ab |
| | | * | *** | *** |

Means within each column followed by the same letters are not significantly different according to the Tukey test.
* and ***: significant at $0.01 < p \leq 0.05$ and $p \leq 0.001$, respectively.

The main and interaction effects of the treatments on the N, P, K, and Mg concentration of the growing medium before seed sowing were found to be significantly different. The initial N concentration (0.81%) of the growing media increased due to the increase in the compost ratio from 0% to 100% at the

start with the rate of 34.57%, 70.37%, 16.05%, 53.09%, 61.73% and 111.11% and for WS, EWS, 2P, E2P, 3P and E3P, respectively. Higher compost ratios produced higher P and K concentrations of the media. The average Ca concentrations of WS, EWS, 2P, E2P, 3P, and E3P were 2.18%, 2.33%, 2.26%, 2.51%, 3.10%, and 2.25% at the start, while the Mg concentration changed between 0.45% and 0.82% (Table 3).

**Table 3.** Macro nutrient concentrations of the growing medium before seed sowing.

| Composts | Compost Ratios in Peat (%) | N (%) | P (%) | K (%) | Ca (%) | Mg (%) |
|---|---|---|---|---|---|---|
| WS | 0 | 0.81 i | 0.12 f | 0.62 e | 2.33 | 0.80 a |
| | 25 | 0.88 hi | 0.22 ef | 0.63 e | 1.92 | 0.82 a |
| | 50 | 0.94 ghi | 0.28 de | 0.63 e | 3.52 | 0.82 a |
| | 75 | 1.07 e–i | 0.36 cd | 0.71 de | 2.85 | 0.81 a |
| | 100 | 1.09 e–i | 0.41 bc | 0.95 b–e | 2.02 | 0.80 a |
| EWS | 0 | 0.81 i | 0.12 f | 0.62 e | 2.33 | 0.80 a |
| | 25 | 1.03 f–i | 0.67 a | 0.77 cde | 1.97 | 0.69 a–d |
| | 50 | 1.10 e–i | 0.69 a | 0.76 cde | 1.84 | 0.76 ab |
| | 75 | 1.42 a–d | 0.72 a | 1.04 bcd | 2.09 | 0.75 abc |
| | 100 | 1.38 b–e | 0.74 a | 1.12 b | 2.17 | 0.66 a–d |
| 2P | 0 | 0.81 i | 0.12 f | 0.62 e | 2.33 | 0.80 a |
| | 25 | 0.80 i | 0.15 f | 0.68 e | 2.73 | 0.82 a |
| | 50 | 0.95 ghi | 0.22 ef | 0.75 cde | 2.32 | 0.78 ab |
| | 75 | 0.95 ghi | 0.24 ef | 0.69 e | 2.66 | 0.77 ab |
| | 100 | 0.94 ghi | 0.37 cd | 0.90 b–e | 2.18 | 0.77 ab |
| E2P | 0 | 0.81 i | 0.12 f | 0.62 e | 2.33 | 0.80 a |
| | 25 | 0.99 f–i | 0.67 a | 0.72 de | 2.08 | 0.68 a–d |
| | 50 | 1.23 d–g | 0.73 a | 0.89 b–e | 1.97 | 0.68 a–d |
| | 75 | 1.28 d–g | 0.76 a | 1.08 bc | 3.31 | 0.72 a–d |
| | 100 | 1.24 c–g | 0.75 a | 1.51 b | 2.32 | 0.77 a–b |
| 3P | 0 | 0.81 i | 0.12 f | 0.62 e | 2.33 | 0.80 a |
| | 25 | 0.94 ghi | 0.31 cde | 0.84 b–e | 3.05 | 0.81 a |
| | 50 | 1.00 f–i | 0.43 bc | 0.89 b–e | 2.66 | 0.79 ab |
| | 75 | 1.08 e–i | 0.39 bcd | 0.89 b–e | 2.52 | 0.79 ab |
| | 100 | 1.31 b–f | 0.50 b | 1.11 b | 3.87 | 0.72 a–d |
| E3P | 0 | 0.81 i | 0.12 f | 0.62 e | 2.33 | 0.80 a |
| | 25 | 1.19 d–h | 0.74 a | 0.81 b–e | 2.46 | 0.45 de |
| | 50 | 1.59 ab | 0.74 a | 0.75 cde | 2.71 | 0.52 b–e |
| | 75 | 1.56 abc | 0.75 a | 1.77 a | 2.57 | 0.49 cde |
| | 100 | 1.71 a | 0.74 a | 1.80 a | 2.69 | 0.64 a–d |
| | | *** | * | ** | ns | *** |

Means within each column followed by the same letters are not significantly different according to the Tukey test. ns, *, ** and ***: nonsignificant, significant at $0.01 < p \le 0.05$, $0.001 < p \le 0.01$ and $p \le 0.001$, respectively.

The type of composts and ratios also affected the Zn, Mn, and Cu concentrations of the growing media at the start of the experiment. The Zn concentration varied between 68.2 and 432.4 mg kg$^{-1}$, the Mg and Cu concentration varied between 107.8–287.8 mg kg$^{-1}$ and 36.6–55.0 mg kg$^{-1}$ before seed sowing (Table 4).

*3.2. Germination Period and Rate*

The number of days from seed sowing until germination was 4.25 days in local peat (0%) and increased in all composts with increasing compost ratios in the growing medium particularly in the enriched treatments. The use of a compost ratio of 25% in the growing medium shortened the number of days compared with other compost ratios, but extended 11.8%, 17.6%, 5.9%, 17.6%, 5.9% and 111.8% in WS, EWS, 2P, E2P, 3P and E3P, respectively, compared to local peat, while the extension rate was 41.2%, 252.9%, 117.6%, 194.1%, 152.9%, and 264.7% for a compost ratio of 100% compared with local peat (Table 5). The germination rate also showed the same tendency and decreased with increasing compost ratios, but the ratio changed dramatically in the enriched growing medium (Table 5).

**Table 4.** Micro nutrient concentrations of the growing medium before seed sowing.

| Composts | Compost Ratios in Peat (%) | Zn (mg kg⁻¹) | Mn (mg kg⁻¹) | Cu (mg kg⁻¹) |
|---|---|---|---|---|
| WS | 0 | 68.2 d | 136.0 def | 49.7 |
|  | 25 | 108.3 bcd | 184.1 b–f | 37.3 |
|  | 50 | 199.1 bcd | 205.0 a–d | 39.1 |
|  | 75 | 179.6 bcd | 206.4 a–d | 39.4 |
|  | 100 | 240.6 bcd | 230.9 abc | 45.9 |
| EWS | 0 | 68.2 d | 136.0 def | 49.7 |
|  | 25 | 123.9 bcd | 108.5 ef | 50.5 |
|  | 50 | 150.8 bcd | 123.4 def | 50.8 |
|  | 75 | 263.8 abc | 135.8 def | 52.1 |
|  | 100 | 228.9 bcd | 152.3 c–f | 49.7 |
| 2P | 0 | 68.2 d | 136.0 def | 49.7 |
|  | 25 | 86.7 cd | 149.5 c–f | 37.3 |
|  | 50 | 88.9 cd | 110.1 ef | 40.0 |
|  | 75 | 88.4 cd | 116.5 ef | 41.3 |
|  | 100 | 108.9 bcd | 107.8 f | 40.2 |
| E2P | 0 | 68.2 d | 136.0 def | 49.7 |
|  | 25 | 140.7 bcd | 154.5 c–f | 41.9 |
|  | 50 | 171.0 bcd | 186.2 b–f | 52.7 |
|  | 75 | 247.5 a–d | 241.0 ab | 52.1 |
|  | 100 | 432.4 a | 287.8 a | 46.8 |
| 3P | 0 | 68.2 d | 136.0 def | 49.7 |
|  | 25 | 137.3 bcd | 194.6 b–e | 36.6 |
|  | 50 | 162.6 bcd | 185.9 b–f | 38.6 |
|  | 75 | 97.7 bcd | 164.9 b–f | 38.5 |
|  | 100 | 147.1 bcd | 160.2 b–f | 39.9 |
| E3P | 0 | 68.2 d | 136.0 def | 49.7 |
|  | 25 | 132.9 bcd | 120.5 def | 55.0 |
|  | 50 | 215.8 bcd | 131.6 def | 46.7 |
|  | 75 | 134.8 bcd | 167.7 b–f | 45.2 |
|  | 100 | 282.2 ab | 168.3 b–f | 45.4 |
|  |  | * | ** | ns |

Means within each column followed by the same letters are not significantly different according to the Tukey test. ns, * and **: nonsignificant, significant at $0.01 < p \le 0.05$ and $0.001 < p \le 0.01$, respectively.

**Table 5.** Effects of composts with local peat on germination period and the rate of *Solanum lycopersicum*.

| Compost Ratios in Peat (%) | Composts | | | | | | |
|---|---|---|---|---|---|---|---|
|  | WS | EWS | 2P | E2P | 3P | E3P | Mean_ratio |
| **Germination Period (day)** | | | | | | | |
| 0 | 4.25 j | 4.25 j | 4.25 j | 4.25 j | 4.25 j | 4.25 j | 4.25 E |
| 25 | 4.75 ij | 5.00 ij | 4.50 j | 5.00 ij | 4.50 j | 9.00 fg | 5.46 D |
| 50 | 6.00 hi | 9.00 fg | 5.25 hj | 9.00 fg | 6.50 h | 12.75 b | 8.08 C |
| 75 | 8.00 g | 11.25 cd | 8.00 g | 11.75 bd | 9.50 ef | 15.50 a | 10.67 A |
| 100 | 6.00 hi | 15.00 a | 9.25 fg | 12.50 bc | 10.75 de | 0.00 k * | 8.92 B |
| Mean_compost | 5.80 D | 8.90 A | 6.25 D | 8.50 AB | 7.10 C | 8.30 B | |
| **Germination Rate (%)** | | | | | | | |
| 0 | 94.92 ab | 94.92 ab | 94.92 ab | 94.92 ab | 94.92 ab | 94.92 ab | 94.92 A |
| 25 | 94.14 ab | 90.23 bc | 91.41 ac | 95.31 ab | 92.58 ab | 92.97 ab | 92.77 AB |
| 50 | 91.18 ac | 94.92 ab | 91.02 ac | 93.36 ab | 94.92 ab | 79.30 e | 90.78 B |
| 75 | 96.88 a | 85.94 cd | 92.58 ab | 81.64 de | 91.02 ac | 70.70 f | 86.46 C |
| 100 | 94.53 ab | 54.02 g | 94.53 ab | 71.88 f | 76.95 ef | 14.45 h | 67.73 D |
| Mean_compost | 94.33 A | 84.01 D | 92.89 AB | 87.42 C | 90.08 BC | 70.47 E | |

* "0" is accepted as germination rates lower than 50%. Means within each column followed by the same letters are not significantly different according to the Tukey test. Capital letters show significant differences in mean values of composts and compost ratios in peat; lowercase letters indicate significant differences in interaction.

### 3.3. Seedling Growth

The effects of the treatments on the lengths of shoots and roots and stem diameter were found to be significantly different (Table 5). The shoot length changed between 16.33 and 4.65 cm. A compost ratio of up to 50% in the growing medium promoted the shoot length, but an increasing compost ratio had an impact on shoot growth excluding the compost ratio of 75% in WS. The shoot length sharply decreased in E2P and E3P. The root length was similar in the treatments, but it decreased by 35% in E3P. The stem diameter also showed similarities to the other measured parameters and decreased in the enriched treatments with an increasing compost ratio (Table 6).

**Table 6.** Effects of treatments on growth.

| Compost Ratios in Peat (%) | Composts | | | | | | |
|---|---|---|---|---|---|---|---|
| | WS | EWS | 2P | E2P | 3P | E3P | Mean$_{ratio}$ |
| | Shoot Length (cm) | | | | | | |
| 0 | 4.95 mn | 4.95 mn | 4.95 mn | 4.95 mn | 4.95 mn | 4.95 mn | 4.95 D |
| 25 | 13.03 d | 15.66 ab | 11.10 eg | 15.13 bc | 14.48 c | 6.43 kl | 12.63 A |
| 50 | 14.38 c | 16.33 a | 10.40 gi | 12.23 de | 14.73 bc | 5.25 mn | 12.22 A |
| 75 | 15.65 ab | 11.19 eg | 9.93 hi | 7.00 jk | 11.83 ef | 4.65 n | 10.04 B |
| 100 | 11.15 eg | 10.89 fh | 7.83 j | 5.85 lm | 9.48 i | 4.74 mn | 8.32 C |
| Mean$_{compost}$ | 11.83 A | 11.80 A | 8.84 C | 9.03C | 11.09 B | 5.20 D | |
| | Root Length (cm) | | | | | | |
| 0 | 7.03 de | 7.03 de | 7.03 de | 7.03 de | 7.03 de | 7.03 de | 7.03 |
| 25 | 7.70 ad | 7.21 ce | 7.78 ad | 7.28 be | 8.00 ac | 6.63 e | 7.43 |
| 50 | 7.28 be | 7.95 ac | 7.83 ad | 7.33 be | 8.03 ac | 4.48 f | 7.15 |
| 75 | 7.25 be | 8.33 a | 8.28 a | 7.38 be | 8.33 a | 3.73 fg | 7.21 |
| 100 | 8.08 ab | 8.40 a | 7.95 ac | 7.38 be | 8.00 ac | 2.93 g | 7.12 |
| Mean$_{compost}$ | 7.47 BC | 7.78 AB | 7.77 AB | 7.28 C | 7.88 A | 4.96 D | |
| | Stem Diameter (mm) | | | | | | |
| 0 | 1.17 jk | 1.17 jk | 1.17 jk | 1.17 jk | 1.17 jk | 1.17 jk | 1.17 E |
| 25 | 2.14 ce | 2.46 a | 1.83 h | 2.46 a | 2.37 ab | 1.33 ij | 2.10 A |
| 50 | 2.41 ab | 2.23 bd | 1.88 fh | 2.05 dg | 2.46 a | 1.01 k | 2.00 B |
| 75 | 2.31 ac | 1.94 eh | 1.84 gh | 1.38 ij | 2.06 dg | 0.77 l | 1.72 C |
| 100 | 2.20 bd | 1.86 fh | 1.52 i | 1.09 k | 2.08 df | 0.79 l | 1.59 D |
| Mean$_{compost}$ | 2.04 A | 1.93 B | 1.65 C | 1.63 C | 2.02 AB | 1.01 D | |

Means within each column followed by the same letters are not significantly different according to the Tukey test. Capital letters show significant differences in mean values of composts and compost ratios in peat; lowercase letters indicate significant differences in interaction.

The treatments affected the dry weights of the roots significantly. Although root dry/fresh weights increased with compost ratios in the growing medium, this tendency did not continue with increasing ratios. Particularly, the values of E2P and E3P with compost ratios of over 25% showed less root growth (Figure 2).

The main and interaction effects of the treatments on shoot growth were also found to be significantly different. The results showed that the highest dry weights were in WS, while the lowest values were determined for the seedlings grown in E3P. Increasing doses of compost ratios of more than 25% and enrichment had negative effects on seedling dry weights (Figure 3).

### 3.4. Chlorophyll Index

The treatments affected the chlorophyll index values (SPAD) significantly. However, there was a slight reduction in WS, 2P, and 3P with increasing compost ratios, whereas the chlorophyll index increased in the enriched compost treatments (Table 7).

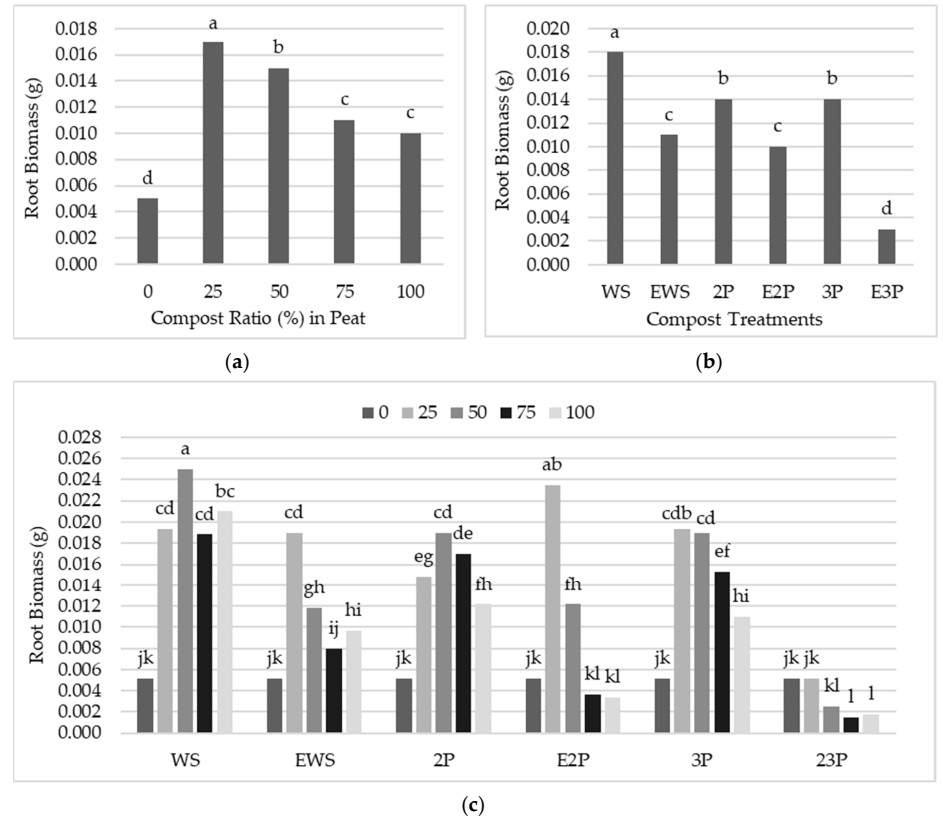

**Figure 2.** Main (**a**,**b**) and interaction (**c**) effects of the treatments on root dry weight. Means within each column followed by the same letters are not significantly different according to the Tukey test.

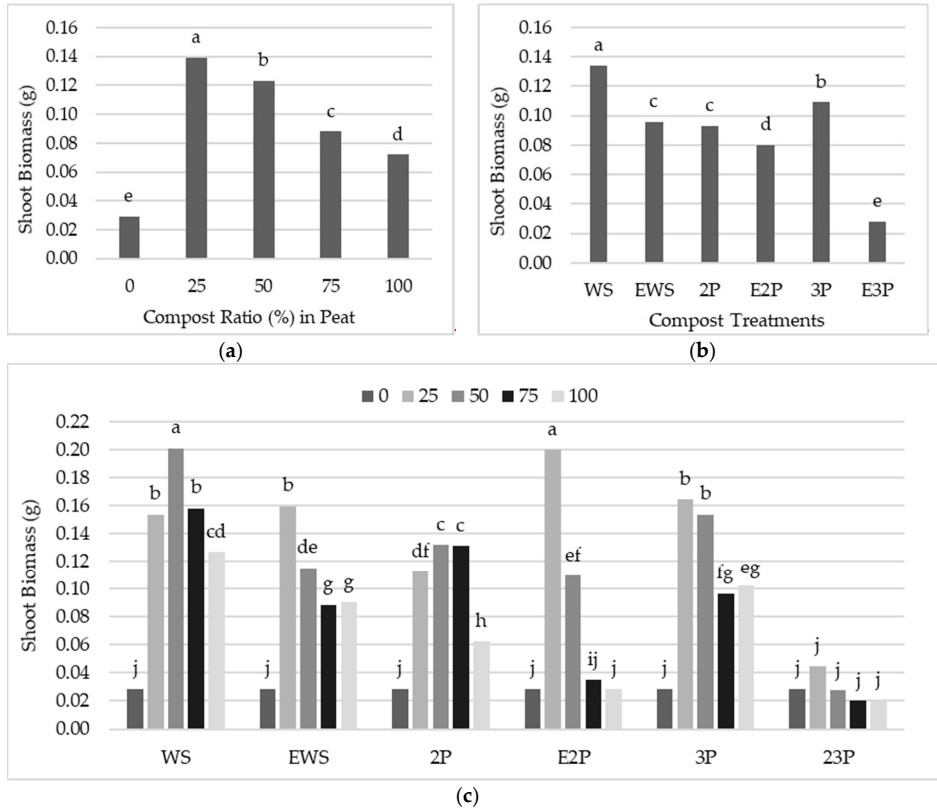

**Figure 3.** Main (**a**,**b**) and interaction (**c**) effects of the treatments on shoot dry weight. Means within each column followed by the same letters are not significantly different according to the Tukey test.

**Table 7.** Effects of the treatments on chlorophyll index values (SPAD).

| Compost Ratios in Peat (%) | Composts | | | | | | |
|---|---|---|---|---|---|---|---|
| | WS | EWS | 2P | E2P | 3P | E3P | Mean$_{ratio}$ |
| 0 | 30.58 d–h | 30.58 d–h | 30.58 d–h | 30.58 d–h | 30.58 d–h | 30.58 d–h | 30.58 BC |
| 25 | 27.45 f–i | 32.69 c–g | 30.94 d–h | 34.81 b–e | 30.03 e–i | 41.63 a | 32.92 A |
| 50 | 26.21 hi | 36.16 ad | 29.18 e–i | 33.23 c–f | 25.43 hi | 39.97 ab | 31.70 ABC |
| 75 | 26.06 hi | 34.33 b–e | 25.84 hi | 34.12 b–e | 25.86 hi | 34.48 b–e | 30.11 C |
| 100 | 24.18 i | 38.69 abc | 26.86 ghi | 34.9 b–e | 29.75 e–i | 37.31 abc | 31.96 AB |
| Mean$_{compost}$ | 26.90 C | 34.49 B | 28.68 C | 33.55 B | 28.33 C | 36.79 A | |

Means within each column followed by the same letters are not significantly different according to the Tukey test. Capital letters show significant differences in mean values of composts and compost ratios in peat; lowercase letters indicate significant differences in interaction.

## 3.5. Leaf Color

The main effects of composts and compost ratios on the "L*" value of leaf color were significant. The lowest "L*" was in the growing medium composed of local peat. Additionally, "L*" was lower in the enriched composts. The compost ratios only affected the "a*" value and the treatments showed significant difference when compared with peat usage. However, the "b*" value was affected by the main and interaction effect of the treatments and the b* values of the enriched composts were lower. The value of "h" changed according to the compost ratios and peat usage and 2P with a compost ratio of 75% gave the lowest hue value. However, "C*" had the same tendency with the "b*" value (Table 8).

**Table 8.** Effects of treatments on leaf color.

| Compost Ratios in Peat (%) | Composts | | | | | | |
|---|---|---|---|---|---|---|---|
| | WS | EWS | 2P | E2P | 3P | E3P | Mean$_{ratio}$ |
| | L* | | | | | | |
| 0 | 28.91 | 28.91 | 28.91 | 28.91 | 28.91 | 28.91 | 28.91 B |
| 25 | 47.92 | 39.63 | 48.18 | 40.61 | 48.30 | 39.59 | 44.04 A |
| 50 | 49.75 | 37.79 | 47.57 | 40.28 | 49.41 | 39.66 | 44.07 A |
| 75 | 49.34 | 38.85 | 48.97 | 43.19 | 51.35 | 40.52 | 45.37 A |
| 100 | 49.04 | 39.14 | 48.39 | 40.14 | 42.91 | 42.29 | 43.65 A |
| Mean$_{compost}$ | 44.99 A | 36.86 B | 44.40 A | 38.63 B | 44.17 A | 38.19 B | |
| | a* | | | | | | |
| 0 | −0.56 | −0.56 | −0.56 | −0.56 | −0.56 | −0.56 | −0.56 A |
| 25 | −14.51 | −15.96 | −12.61 | −15.60 | −17.43 | −15.33 | −15.24 B |
| 50 | −17.15 | −16.73 | −16.17 | −16.52 | −17.69 | −15.09 | −16.56 B |
| 75 | −18.24 | −15.31 | −15.78 | −17.12 | −18.03 | −16.34 | −16.80 B |
| 100 | −16.89 | −15.41 | −15.38 | −15.86 | −15.47 | −14.07 | −15.51 B |
| Mean$_{compost}$ | −13.47 | −12.79 | −12.10 | −13.13 | −13.84 | −12.28 | |
| | b* | | | | | | |
| 0 | 8.38 h | 8.38 h | 8.38 h | 8.38 h | 8.38 h | 8.38 h | 8.38 D |
| 25 | 28.86 a–f | 23.47 efg | 24.60 c–g | 21.52 g | 29.82 a–e | 21.05 g | 24.89 BC |
| 50 | 31.00 abc | 20.64 g | 32.38 ab | 22.22 fg | 33.76 a | 20.56 g | 26.76 AB |
| 75 | 31.13 abc | 20.20 g | 33.37 a | 25.42 b–g | 34.22 a | 21.84 fg | 27.70 A |
| 100 | 32.26 ab | 20.14 g | 30.80 a–d | 21.44 g | 23.83 d–g | 19.09 g | 24.59 C |
| Mean$_{compost}$ | 26.32 A | 18.57 B | 25.90 A | 19.80 B | 26.00 A | 18.18 B | |

**Table 8.** *Cont.*

| Compost Ratios in Peat (%) | Composts | | | | | | |
|---|---|---|---|---|---|---|---|
| | WS | EWS | 2P | E2P | 3P | E3P | Mean_ratio |
| | | | | $h°$ | | | |
| 0 | 173.75 | 173.75 | 173.75 | 173.75 | 173.75 | 173.75 | 173.75 A |
| 25 | 116.71 | 124.30 | 117.08 | 125.99 | 120.31 | 126.07 | 121.74 B |
| 50 | 119.04 | 129.08 | 116.53 | 126.62 | 117.68 | 126.41 | 122.56 B |
| 75 | 120.36 | 127.19 | 115.41 | 123.99 | 117.77 | 126.79 | 121.92 B |
| 100 | 117.71 | 127.45 | 116.44 | 126.51 | 123.18 | 126.44 | 122.96 B |
| Mean_compost | 129.52 | 136.54 | 127.84 | 135.37 | 130.54 | 135.89 | |
| | | | | $C*$ | | | |
| 0 | 9.45 h | 9.45 h | 9.45 h | 9.45 h | 9.45 h | 9.45 h | 9.45 C |
| 25 | 32.30 a–e | 28.39 c–g | 27.64 d–g | 26.58 efg | 34.54 abc | 26.04 efg | 29.25 B |
| 50 | 35.43 ab | 26.59 efg | 36.19 ab | 27.69 d–g | 38.13 a | 25.53 efg | 31.59 A |
| 75 | 36.08 ab | 25.35 fg | 36.94 ab | 30.65 b–f | 38.69 a | 27.27 efg | 32.50 A |
| 100 | 36.45 ab | 25.36 fg | 34.44 a–d | 26.67 efg | 28.43 c–g | 23.76 g | 29.18 B |
| Mean_compost | 29.94 A | 23.03 B | 28.93A | 24.21 B | 29.85 A | 22.41 B | |

Means within each column followed by the same letters are not significantly different according to the Tukey test. Capital letters show significant differences in mean values of composts and compost ratios in peat; lowercase letters indicate significant differences in interaction.

### 3.6. Nutrient Concentration

Individual effects of composts and compost ratio with local peat and their interactions on the N and P concentrations of the seedlings showed a similar effect. Based on the interactions, both nutrient concentrations containing enriched composts with compost ratios of 50%, 75%, and 100% for EWS and E2P and with compost ratios of 25%, 50%, and 75% for E3P were higher than those of the composts without enrichment. The mean plant nutrient concentrations of compost rates significantly varied from 2.71% (a compost ratio of 25%) to 3.54% (a compost ratio of 75%) for N, and from 0.15% (0%) to 0.76% for P (Table 9). As for the plant Ca concentration obtained from composts × compost ratio interactions, increasing the compost ratios resulted in a decrease of Ca in plant tissue. This result implies that 100% local peat as seedling substrate had the highest Ca concentration. These results can also be obtained from the compost ratio comparison. The mean values showed that the Ca concentrations obtained from E3P were higher than those obtained from other composts. The plant Mg concentrations showed a similar tendency to Ca. Namely, except for E3P with a compost ratio of 25%, all the other plant Mg concentrations measured from the plugs with 100% local peat were higher. Furthermore, higher compost ratios generally led to a decrease in the plant Mg concentrations. The same trend was recorded from the means of compost ratios. While the lowest Mg concentrations were determined from 2P, there was not a significant variation among the means of the other composts (Table 9).

**Table 9.** Effects of the treatments on macro element concentrations of leaves.

| Compost Ratios in Peat (%) | Composts | | | | | | |
|---|---|---|---|---|---|---|---|
| | WS | EWS | 2P | E2P | 3P | E3P | Mean_ratio |
| | | | | N (%) | | | |
| 0 | 3.05 c–f | 3.05 c–f | 3.05 c–f | 3.05 c–f | 3.05 c–f | 3.05 c–f | 3.05 AB |
| 25 | 1.73 ef | 3.46 b–e | 2.01 ef | 2.98 c–f | 1.90 ef | 4.19 a–d | 2.71 B |
| 50 | 1.90 ef | 4.79 abc | 1.27 f | 4.81 abc | 1.87 ef | 4.41 a–d | 3.17 AB |
| 75 | 2.12 ef | 5.02 ab | 1.27 f | 5.21 ab | 2.22 ef | 5.39 a | 3.54 A |
| 100 | 2.21 ef | 4.40 a–d | 2.33 ef | 5.33 a | 2.70 def | 3.48 b–e | 3.41 A |
| Mean_compost | 2.20 B | 4.14 A | 1.98 B | 4.28 A | 2.35 B | 4.10 A | |

<div align="center">**Table 9.** *Cont.*</div>

| Compost Ratios in Peat (%) | Composts | | | | | | |
|---|---|---|---|---|---|---|---|
| | WS | EWS | 2P | E2P | 3P | E3P | Mean$_{ratio}$ |
| | **P (%)** | | | | | | |
| 0 | 0.15 g | 0.15 g | 0.15 g | 0.15 g | 0.15 g | 0.15 g | 0.15 D |
| 25 | 0.28 fg | 0.71 bc | 0.20 g | 0.69 bc | 0.32 fg | 0.84 ab | 0.50 C |
| 50 | 0.41 d–g | 0.83 ab | 0.37 efg | 0.84 ab | 0.59 b–e | 0.83 ab | 0.64 B |
| 75 | 0.68 bcd | 0.84 ab | 0.51 c–f | 0.78 ab | 0.71 bc | 1.05 a | 0.76 A |
| 100 | 0.61 b–e | 0.81 ab | 0.63 b–e | 0.81 ab | 0.75 bc | 0.69 bc | 0.72 AB |
| Mean$_{compost}$ | 0.42 BC | 0.67 A | 0.37 C | 0.65 A | 0.50 B | 0.71 A | |
| | **Ca (%)** | | | | | | |
| 0 | 6.70 a | 6.70 a | 6.70 a | 6.70 a | 6.70 a | 6.70 a | 6.70 A |
| 25 | 2.69 b–g | 2.88 b–f | 3.18 bcd | 2.86 b–f | 3.05b–e | 3.36 bc | 3.00 B |
| 50 | 2.49 c–g | 2.36 d–g | 2.68 b–g | 2.25 d–g | 2.49 c–g | 2.84 b–f | 2.52 C |
| 75 | 2.10 efg | 1.87g | 2.28 d–g | 2.39 d–g | 2.32 d–g | 2.81 b–g | 2.30 C |
| 100 | 2.02 fg | 1.96fg | 2.77 b–g | 2.84 b–f | 2.26 d–g | 3.56 b | 2.57 C |
| Mean$_{compost}$ | 3.20 BC | 3.15 C | 3.52 B | 3.41 BC | 3.36 BC | 3.85 A | |
| | **Mg (%)** | | | | | | |
| 0 | 0.93 b | 0.93 b | 0.93 b | 0.93 b | 0.93 b | 0.93 b | 0.93 A |
| 25 | 0.78 bc | 0.78 bc | 0.7 0 bc | 0.76 bcd | 0.75 bcd | 1.20 a | 0.84 B |
| 50 | 0.73bcd | 0.76 bcd | 0.71 bcd | 0.76 bcd | 0.73 bcd | 0.76 bc | 0.74 C |
| 75 | 0.70 b–e | 0.77 bc | 0.67 cde | 0.78 bc | 0.72 bcd | 0.63 cde | 0.71 C |
| 100 | 0.57 cde | 0.77 bc | 0.45 e | 0.58 cde | 0.72 bcd | 0.51 de | 0.60 D |
| Mean$_{compost}$ | 0.74 AB | 0.80 A | 0.70 B | 0.76 AB | 0.77 AB | 0.81 A | |

Means within each column followed by the same letters are not significantly different according to the Tukey test. Capital letters show significant differences in mean values of composts and compost ratios in peat; lowercase letters indicate significant differences in interaction.

The plant Zn concentrations were significantly affected by individual factors and their interactions (Table 10). The Zn concentrations increased with increasing compost ratios. The Zn concentrations of the seedlings grown on the enriched composts were usually higher than those of the other composts without enrichment and the highest values were measured from E3P with a compost ratio of 75% and E3P with a compost ratio of 100% with the values of 325 and 226 mg kg$^{-1}$ Zn in seedling tissue. Compared to the control (0%), the plant Zn concentrations showed more than threefold increment with increasing compost ratios up to 75%. The means of composts showed that Zn levels determined from the enriched compost were higher than those obtained from non-enriched composts. The highest Zn concentration was measured from the plants growing on E3P. The individual effects of composts and compost ratio showed a significant effect on the Mn and Cu concentrations (Table 10). While the seedling Mn concentrations increased with the compost ratio, the plant Cu concentrations decreased, but no significant differences were observed among compost ratios between 25% and 100%. The results show that the enriched composts seemed to be more effective than the non-enriched composts on the plant Mn concentrations. Additionally, WS was statistically in the same group. The Mn concentrations obtained from 2P and 3P substrates had the lowest values. Similarly, the plant Cu concentrations measured from the enriched composts were higher than those measured from the non-enriched composts and the highest Cu concentration was determined from the plant grown on E3P. WS had the lowest effect on the plant Cu concentration.

**Table 10.** Effects of the treatments on micro element concentrations of leaves.

| Compost Ratios in Peat (%) | Composts | | | | | | |
|---|---|---|---|---|---|---|---|
| | WS | EWS | 2P | E2P | 3P | E3P | Mean$_{ratio}$ |
| | Zn (mg kg$^{-1}$) | | | | | | |
| 0 | 44 g | 44 g | 44 g | 44 g | 44 g | 44 g | 44 C |
| 25 | 49 fg | 91 c–g | 76 d–g | 95 c–g | 67 efg | 103 c–g | 80 B |
| 50 | 67 efg | 143 b–g | 67 efg | 160 b–f | 86 c–g | 188 bc | 118 AB |
| 75 | 84 c–g | 170 b–e | 64 efg | 155 b–f | 100 c–g | 325 a | 150 A |
| 100 | 101 c–g | 150 b–g | 100 c–g | 186 bcd | 125 b–g | 226 ab | 148 A |
| Mean$_{compost}$ | 69 D | 120 BC | 70 D | 128 B | 85 CD | 177 A | |
| | Mn (mg kg$^{-1}$) | | | | | | |
| 0 | 32 | 32 | 32 | 32 | 32 | 32 | 32 C |
| 25 | 12 | 15 | 14 | 18 | 18 | 33 | 18 C |
| 50 | 10 | 50 | 15 | 52 | 26 | 60 | 36 C |
| 75 | 70 | 83 | 40 | 62 | 52 | 89 | 66 B |
| 100 | 114 | 104 | 90 | 102 | 69 | 76 | 93 A |
| Mean$_{compost}$ | 49 AB | 57 A | 38 B | 53 A | 39 B | 58 A | |
| | Cu (mg kg$^{-1}$) | | | | | | |
| 0 | 28 | 28 | 28 | 28 | 28 | 28 | 28 A |
| 25 | 9 | 14 | 12 | 12 | 8 | 21 | 13 B |
| 50 | 7 | 15 | 11 | 19 | 11 | 24 | 14 B |
| 75 | 7 | 14 | 9 | 16 | 16 | 22 | 14 B |
| 100 | 6 | 14 | 7 | 20 | 14 | 24 | 14 B |
| Mean$_{compost}$ | 11 C | 17 AB | 13 BC | 19 AB | 15 ABC | 24 A | |

Means within each column followed by the same letters are not significantly different according to the Tukey test. Capital letters show significant differences in mean values of composts and compost ratios in peat; lowercase letters indicate significant differences in interaction.

## 4. Discussion

Seedlings are grown in a limited volume of containers, however, materials and rates utilized in formulations of growing medium affect the physical, chemical and/or biological properties of medium [26], which is also directly linked with seedling quality. Growing medium provides physical support, aeration, supply of water, and nutrients [27]. In our experiments, the enrichment of the growing medium and also increasing the compost ratio increased organic matter content, electrical conductivity, and macro and micro element concentrations. The origin of compost also affects the nutritional features of growing medium. Furthermore, olive oil processing wastes are rich in nutrients with a higher electrical conductivity [28,29]. Although there were slight changes in organic matter content before planting, P, Ca, Mg, Zn, and Mn decreased during the seedling growth due to plant consumption. However, the increase in N was most probably due to the ongoing mineralization affected by the composition and the characteristics of the material, temperature, and water content [30].

The germination rate changed between 14.45% and 96.88% and decreased by the enrichment of the growing medium in particular in EWS and E2P when the compost ratio was 75% and over, while the germination rate declined in E3P after a compost ratio of 50% and with the increasing compost ratio in the growing medium. However, the germination period also lasted longer with the enrichment of the growing medium and increasing compost ratios. Sánchez–Monedero et al. [31] also reported a lower germination rate and a delay in seedling emergence when the relative proportion of the compost increased in the growing medium, leading to higher EC. The rate and duration of germination are affected by the physical and chemical properties of the growing medium, the rate of ingredients, the requirement of crop species, and crop management including irrigation, fertigation, and the use of beneficial microorganisms as well as environmental conditions [32].

In terms of germination rate, two composts made from olive pomace waste and green waste were used as growing medium components at four ratios (20%, 45%, 70%, 90%, *v/v*) and compost made of green waste with ratio 20% and 45% and olive pomace waste with ratio of 20% showed the best performances [29]. Perez-Murcia et al. [33] tested the addition of increasing quantities of composted sewage sludge to peat (0%, 15%, 30%, and 50%, *v/v*), and increasing sewage sludge treatments (especially 30% and 50%) reduced the germination of lettuce and broccoli, but in cauliflower seedlings, an increment of germination was observed for the 15% and 30% treatments compared with the control. A compost ratio of 25% for composted rose oil processing [34] and for olive oil production wastes [35] was found appropriate in terms of the rate and duration of germination for organic tomato seedling production which is in harmony with our results.

Healthy seedling growth is a prerequisite for the success of crop production [36]. The shoot length was lower in compost ratios over 50% excluding WS, which reacted to over 75%. Shoot length and stem diameter decreased by the enrichment of the growing medium over 50% compost rate in EWS and E2P. The longest root lengths were also affected by the enrichment of medium excluding WS and EWS which could be also be related to the washing process. The development of shoot, root, and stem was the poorest in E3P. The nutrient contents of the growing media were higher in the ones with higher compost ratios and the enriched ones (Table 2), but the EC values were also high in those ones. The highest average EC value was in E3P treatment, resulting in the poorest shoot, root and stem development.

Tomato is moderately sensitive to salinity and salinity threshold of tomatoes is 2.5 dS m$^{-1}$ [37]. Increasing salinity in the rhizosphere restricts root cell growth and increases root lesion, resulting in a reduction in root elongation rate and lateral root growth. Additionally, a reduction in photosynthesis and tissue expansion and the inhibition of cell division affect leaf and shoot growth [38]. Maggio et al. [39] found that high EC (approx. 9.6 dS m$^{-1}$) caused a sharp increase in the values of root and shoot abscisic acid (ABA), which coincided with the reduction of stomatal resistance to ABA, a different partitioning of Na ions between young and mature leaves, and the increase of root to shoot ratio [39]. In our experiment, morphological measurements (a decrease in shoot length, stem diameter, shoot and root biomass with an increasing compost ratio and enrichment process, poor growth particularly in under E2P and E3P) and SPAD readings, which showed the greenness or the relative chlorophyll concentration of leaves and the highest root to shoot dry matter ratio (in E3P), confirm the effect of salt stress on the seedlings.

The highest plant dry weights were measured from the plants grown on the media with compost ratios of 50% and 25% for WS and E2P, respectively. The variation of the results could be explained in terms of the chemical composition of the composts [40–42]. However, some other properties such as humic and fulvic acid and some other hormones like substances may also have positive effects on plant growth, and thus dry weight [43]. The decrease of dry weight with an increase higher than 50% in compost ratio either enriched or not might be due to the toxicity of some fenolic compounds on plant growth [44,45]. In order to prevent the toxic effect of WS, it was reported to follow the changes occurring in phenols and biotoxicity during composting. Moreover, Zenjari et al. [46] indicated that toxicity disappeared after 2 months of composting. Many studies conducted with different plants grown on different composting materials proposed rates of WS in composting between 25% and 67% [31,47]. The enrichment of 2P (E2P) with P and Ca due to different materials, especially rock phosphate, may have a positive effect on plant growth and dry weight.

The results show that all the composts, either enriched or not, and compost ratios had significantly different effects on most of the plant nutrient concentrations. If a general evaluation is made for the plant N, P, and Zn concentrations, it can be clearly seen that these nutrient concentrations in plants grown on the enriched composts were higher than the non-enriched composts. A number of studies showed that pre-mixing rock phosphate with agro-wastes followed by composting increased the P availability to plants [48–51]. Local peat seems to be the best medium in terms of the plant Ca and Mg concentrations. However, it is quite clear that the dilution effect played a very important role especially

for Ca, as dry weights obtained from 100% local peat containing plug were quite low when compared to most of the media. It is well-documented in the literature that nutrients are diluted in plant tissues with plant growth and concentrated with growth retention [41,52].

Among the tested compost ratios, a ratio of 25% was found appropriate in most of the measured properties. However, compost ratios could be increased by up to 50% in the case of water sludge use. Previous research results also propose a rate starting from 25% up to 67% in different crops (such as poinsettia with olive mill wastes [53]; tomato with municipal solid waste compost [47]; broccoli, onion, and tomato with sweet sorghum bagasse, pine bark, and either urea or brewery sludge [31]; lettuce, chard, broccoli, and coriander with exhausted grape marc and cattle or poultry manure [54]). The chemical and physical properties of compost affect the compost ratio in the growing medium [47] and nitrogen has the greatest effect on transplant growth [55]. In our experiment, the higher EC level of the growing medium when enriched and/or included higher compost ratio affected plant growth starting from the seed germination stage. These results are in harmony with the results of our experiments conducted with composts containing rose oil processing wastes [34] and olive oil production wastes [35].

Peat is the most common substrate in seedling production. Although peat-based growing media are allowed in organic production, peat substitution in plant nursery activity and, in particular, in organic seedling production is a debated issue [56] since peat utilization contradicts numerous fundamental principles of organic agriculture. EGTOP (Expert Group for Technical Advice on Organic Production) advises that its use in growing media should be limited to a maximum of 80% by volume, as normally 20%–30% of peat by volume in growing media for professional use could be replaced by compost [57]. Our results showed that composts based on olive mill wastes and olive oil wastewater sludge could be used in the growing medium of vegetable seedlings and there is no need to enrich the medium, which results in a much higher electrical conductivity and higher costs.

Future studies should focus on the enrichment of composts with the effective microorganisms to improve soil fertility and facilitate the nutrient uptake from the soil.

## 5. Conclusions

In conclusion, the composts obtained from two-phase and three-phase olive mill solid wastes and olive oil wastewater sludge can be used without any need of enrichment and a ratio of 25% was found appropriate in most of the measured properties. However, compost ratios could be increased by up to 50% in the case of water sludge compost use.

**Author Contributions:** All the authors contributed to the study conception and design. Material preparation, data collection and analysis were performed by Y.T., K.E., G.B.Ö., İ.E., N.V. and Ö.M. The manuscript was written by Y.T., K.E., G.B.Ö. and İ.E. All the authors read and approved the final manuscript. All authors have read and agreed to the published version of the manuscript.

**Funding:** This research was carried out in the framework of the project "Developing of Input Production Methods for Utilization in Organic Plant Production", which was approved by The Scientific and Technological Research Council of Turkey (TUBİTAK) with project number 11-G−055.

**Acknowledgments:** The authors are grateful to Gulay Beşirli for organic seed supply.

**Conflicts of Interest:** The authors declare no conflict of interest.

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
