# Peer review of "Utilization of Olive Oil Processing Waste Composts in Organic Tomato Seedling Production"

_agronomy, doi:10.3390/agronomy10060797_

Round 1
Reviewer 1 Report
Keep up this kind of good work related to composting.
Author Response
We would like to thank experts for their invaluable comments. Minor revisions have been highlighted in yellow on the manuscript.

Reviewer 2 Report
The work describes an interesting and innovative topic.
The methodologies are well described and the discussion of the results clearly highlights the innovation compared to previous studies.
Author Response

(The authors gave the same response as above.)

Reviewer 3 Report
I have no comments. The authors processed my comments from the first revision.
Author Response
We would like to thank experts for their invaluable comments. Minor revisions have been highlighted in yellow on the manuscript.

This manuscript is a resubmission of an earlier submission. The following is a list of the peer review reports and author responses from that submission.
Round 1
Reviewer 1 Report
The quality of English and content in the Method section dropped compared to the rest.
If possible - please add more information on the composting process (eg temperature, quantitative info etc) in the Method section.
Please address the comments made on the attached pdf.

Author Response
We would like to thank referees for their invaluable comments. All the corrections and additions are listed in attached file and have been marked on the manuscript.
Sincerely,
Yuksel Tuzel, pHD
Ege University,
Faculty of Agriculture
Department of Horticulture,
35100, Izmir, Turkey
yuksel.tuzel@ege.edu.tr

Reviewer 2 Report
- English should be checked, especially the use of commas and brackets
- affiliations should be simplified: 1 =3; 2 = 4
- in the Discussion lines 283 - 285 must be eliminated
- the tables are not immediate readable, it would be advisable to align the numbers as follows:
| 145 | ab |
| 12 | fg |
| 48 | c |
Author Response

(The authors gave the same response as above.)

Reviewer 3 Report
This manuscript focused on evaluating the effects of composting (with and without enrichment) obtained from olive processing on the production of tomato plants of varying rates in growth media. In Turkey, this is a topical issue, other olive-processing countries can use the data. The methodology of the research and the results is described in great detail. In the methodology I miss photos from research. I recommend converting some tables into a graph and inserting the data into an appendix. The graphs are clearer and the reader is better oriented in the results. Formal comments: tons is not SI unit (line 36), ml is written mL (line 137 and 141) – edit in the text. The discussion is extensive and the authors refer to a number of publications. The conclusion is brief. The authors refer to relevant literature. The content of the manuscript is on the good level.
Author Response

(The authors gave the same response as above.)

Reviewer 4 Report
You have a great concept for recycling waste into resources in the olive oil industry. However, there is more work to be done and I encourage you to look at the comments in the attached pdf file critically to help improve your work. Wishing you the best.
The results needs to be arranged better for easy comparisons based on the research questions.
the discussion needs to be detailed to tell what worked best or not and why with supporting evidence from literature or from your results.
the conclusion do not support your research question and needs to be written to be inline with the research question.

Author Response

(The authors gave the same response as above.)
